# Climate Change, Weather, Housing Precarity, and Homelessness: A Systematic Review of Reviews

**DOI:** 10.3390/ijerph18115812

**Published:** 2021-05-28

**Authors:** Mariya Bezgrebelna, Kwame McKenzie, Samantha Wells, Arun Ravindran, Michael Kral, Julia Christensen, Vicky Stergiopoulos, Stephen Gaetz, Sean A. Kidd

**Affiliations:** 1Department of Psychology, York University, Toronto, ON M3J 1P3, Canada; mariya.bezgrebelna@camh.ca; 2Department of Psychiatry, University of Toronto, Toronto, ON M5T 1R8, Canada; kwame@wellesleyinstitute.com (K.M.); samantha.wells@camh.ca (S.W.); arun.ravindran@camh.ca (A.R.); vicky.stergiopoulos@camh.ca (V.S.); 3Centre for Addiction and Mental Health, Institute for Mental Health Policy Research, Toronto, ON M5T 1R8, Canada; 4Centre for Addiction and Mental Health, Campbell Family Mental Health Research Institute, Toronto, ON M5T 1R8, Canada; 5Dalla Lana School of Public Health, University of Toronto, Toronto, ON M5T 3M7, Canada; 6Department of Epidemiology and Biostatistics, Schulich School of Medicine and Dentistry, Western University, London, ON N6A 5C1, Canada; 7School of Psychology, Deakin University, Burwood, VIC 3125, Australia; 8Department of Social Work, Wayne State University, Detroit, MI 48202, USA; michael.kral@wayne.edu; 9Department of Geography, Memorial University Newfoundland, St. John’s, NL A1C 5S7, Canada; jchristensen@mun.ca; 10Faculty of Education, York University, Toronto, ON M3J 1P3, Canada; SGaetz@edu.yorku.ca

**Keywords:** homeless, housing, climate, weather, review, health

## Abstract

This systematic review of reviews was conducted to examine housing precarity and homelessness in relation to climate change and weather extremes internationally. In a thematic analysis of 15 reviews (5 systematic and 10 non-systematic), the following themes emerged: risk factors for homelessness/housing precarity, temperature extremes, health concerns, structural factors, natural disasters, and housing. First, an increased risk of homelessness has been found for people who are vulnerably housed and populations in lower socio-economic positions due to energy insecurity and climate change-induced natural hazards. Second, homeless/vulnerably-housed populations are disproportionately exposed to climatic events (temperature extremes and natural disasters). Third, the physical and mental health of homeless/vulnerably-housed populations is projected to be impacted by weather extremes and climate change. Fourth, while green infrastructure may have positive effects for homeless/vulnerably-housed populations, housing remains a major concern in urban environments. Finally, structural changes must be implemented. Recommendations for addressing the impact of climate change on homelessness and housing precarity were generated, including interventions focusing on homelessness/housing precarity and reducing the effects of weather extremes, improved housing and urban planning, and further research on homelessness/housing precarity and climate change. To further enhance the impact of these initiatives, we suggest employing the Human Rights-Based Approach (HRBA).

## 1. Introduction

Climate refers to the probability of specific weather characteristics (e.g., temperature, wind, rain) occurring at a given location over a given period [1]. Climate change, in turn, is defined as “a change of climate which is attributed directly or indirectly to human activity that alters the composition of the global atmosphere and which is in addition to climate variability observed over comparable time periods” [2] (p. 7). Global climate change has a wide range of significant, direct, negative consequences for human populations. These impacts include illness morbidity and mortality related to heat exposure and extreme weather events alongside systems effects upon agriculture, trade, labor, energy supply and demand, social interactions (violence, disrupted institutions), population growth, structure, and movement [1].

An equity-based perspective on how climate change impacts individuals and populations is fundamental to improving and nuancing our understanding of the health and social effects of changing environments. While there are biological vulnerabilities as a function of age—for example, with unique risks in utero, early childhood, and old age—differential effects related to social vulnerabilities are prominent. Social vulnerability refers to the sensitivity of a population to natural hazards and the ability to respond to the impacts of those hazards [3]. Socioeconomic disadvantage, race, and gender are key to social vulnerability–climate intersections. Impoverished and marginalized individuals and communities globally are more exposed to weather hazards that impact food security and health, threats to safety due to climate-linked conflict and social breakdown, and displacement and migration [4]. These threats apply to both exposure and the resources with which a response can be generated and manifest as a function of sociopolitical and geographic factors. Furthermore, just as social vulnerability interacts with climate change, it also interacts with emerging climate change mitigation policies that can impose more significant burdens upon impoverished populations [5].

Housing precarity and homelessness are critical considerations in the context of the impacts of climate change on human populations. Housing precarity and homelessness represent a multi-dimensional construct ranging from insecure or inadequate housing to a total lack of shelter [6]. A recent scoping review on the impacts of climate and weather on the health of homeless populations found that weather extremes exacerbate vulnerabilities such as chronic illnesses, physical exposure, and stigmatization [7]. This review also highlighted the dearth of evidence in this area despite the global scale and significance of the population health risks involved. A large and diverse literature exists touching on domains relating to housing, homelessness, housing vulnerability, and the impact of climate change; however, such research has not been synthesized and thematically analyzed.

To address this gap in the field, the present systematic review of reviews was conducted to examine broader issues of poverty, housing precarity, and homelessness as they relate to climate change and weather extremes internationally. This systematic review method, while less common than reviews of original reports, is increasingly used when policy/systems implications are a primary objective and where the review topic is very broad [8]. The research question for this review was the following: What are the impacts and implications of climate change and weather for vulnerably housed and homeless populations globally as evidenced in reviews of the academic literature? The information generated in the review was intended to shed light on the relevant factors for understanding the interaction between homelessness/housing vulnerability and climate change. This review was also intended to identify existing strategies and interventions that can inform policymakers, service providers, funding agencies, and other relevant actors, and to provide guidance for future research.

## 2. Methods

A systematic review of literature reviews was employed, as this method facilitates the articulation of policy-relevant observations and recommendations in broad research areas. The review methods align with those articulated by Pollock et al., 2017 [9]. First, a question was developed to give the review focus. Second, a systematic search of five electronic databases was conducted up to September 2020 by two authors (MB and SAK). The following combination of search terms was used in four of the databases (PubMed, PsycINFO, Web of Science, and Google Scholar): (poverty OR impoverish*) AND (climate OR weather) AND (hous* OR homeless*) AND (review OR synthesis). In the fifth database, Scopus, a limitation was used (LIMIT-TO (DOCTYPE, “re”)) instead of (review OR synthesis) keywords. For Google Scholar, the search stopped following 100 consecutive unsuccessful hits after the last successful hit in the produced list of identified documents. Additional sources were also identified through bibliography scans of included references.

Third, the inclusion criteria were developed prior to abstract screening and included (a) reviews, both systematic and non-systematic (anticipating potential gaps in evidence); (b) works addressing the impacts of climate change or weather on homeless or vulnerably housed populations; and (c) works that were available in English. The reviews were screened based on their abstract and/or partial paper review, followed by full paper reviews. The same procedure was followed for papers retrieved from the reference lists. The first author (MB) conducted an initial screening, with all uncertainties being assessed by and discussed with another author (SAK). Further, these two authors assessed the quality of all included reviews using the critical appraisal checklist—a tool developed by the umbrella review methodology working group [10]. The decision to include a review as systematic was based on meeting the following criteria from the checklist: a clear research question, clearly defined inclusion criteria, clearly identified search strategy, adequate sources used for the search, and clearly defined methods used to combine studies. The authors compared their quality assessment results, discussed the differences, and reached a consensus. Reviews that met the selected criteria were included in this review of reviews as systematic, whereas the reviews that did not meet the criteria were included as non-systematic. The evidence generated was synthesized via thematic analysis [11].

## 3. Results

### 3.1. Search Results and Reviews Included

The systematic literature search and “snowballing” (the scanning of included records’ bibliographies and screening of potentially relevant reviews) yielded 503 records. Of these, 23 records were eligible for full-text assessment, following which eight records were excluded. Consequently, 15 reviews were included in this review of reviews (Figure 1). The selected articles included 5 systematic and 10 non-systematic reviews. They were published between the years 2004 and 2019. Most of the reviews had a global or North American focus.

The results of our review of reviews are presented according to the six key themes identified: (1) risk factors for homelessness, (2) heat and cold, (3) health, (4) structural factors, (5) natural disasters, and (6) housing (Table 1). Where available, the presentation of data is organized based on their source (systematic vs non-systematic reviews) and policy recommendations and future research suggestions.

### 3.2. Risk Factors for Homelessness

The risk factors for homelessness as they relate to climate change were assessed in four reviews: one systematic [19] and three non-systematic [20,21,23]. The systematic review focused on energy insecurity and mostly drew on research from the U.S. and the Global North. The non-systematic reviews discussed climate change itself as a factor: two of the reviews discussed it in global terms [20,21], while one focused specifically on the U.S. [23]. 

#### 3.2.1. Energy Insecurity and Climate Change

Energy insecurity tends to be a problem for precariously-housed populations, especially for those living in subsidized and manufactured housing [19]. The inability to afford increased rent and bills associated with heating and cooling may lead to utility shut-offs, which often result in evictions and loss of housing [19]. Thus, energy insecurity is an existing risk factor for homelessness, which is further exacerbated by climate change: “those already experiencing energy insecurity are most affected by climate events” [19] (p. 14) due to their inability to prepare for, withstand, and recover from extreme temperatures and natural disasters.

Climate change was described as a prominent risk factor for further exacerbating homelessness and deteriorating housing conditions. Climate change-induced homelessness is linked to a lack of available resources, limiting the ability of precariously-housed people to respond to environmental disasters and to recover from the damages suffered [23]. It is also projected that environmental changes will prompt migration from rural to urban areas, with a large proportion of such environmental refugees living in informal settlements, further exposing them to environmental disasters [20,21]. Finally, the ability to migrate is also an economically determined factor and, despite the environmental damage, “the poorest of the poor are often unable to migrate and may end up trapped in environmentally degraded areas” [21] (p. 546). Thus, climate change may expose an increasing number of individuals to the risk of homelessness.

#### 3.2.2. Policy Recommendations and Further Research

In addition to providing evidence on factors increasing risk for homelessness, the reviews also offered policy recommendations to address these risk factors. To address energy insecurity, it was suggested that policies targeting structural issues are needed: “policies that address food insecurity, housing insecurity, structural and institutional racism, neighborhood segregation, education inequality, income inequality, and so many other social issues will also affect energy insecurity and together impact population health” [19] (p. 14). In relation to climate change as a risk factor, policy recommendations focus more on ensuring safe living conditions via such methods as urban planning and regulations relating to housing availability and conditions, land use, and infrastructure [20]. However, specific guidance on how these methods are to be implemented was not provided. To better understand and mitigate the climate implications for homelessness, reviews suggested that research is needed on the relationship between climate change and energy insecurity needs: “Research should incorporate and explore the detrimental implications of climate change when evaluating energy insecurity to better prepare for future climate scenarios” [19] (p. 14). Although the developing world tends to be disproportionately affected by climate change, the possible income reductions in the developed world should be also explored [21].

### 3.3. Heat and Cold

The influence of extreme heat events on homeless populations was explored in four reviews: one systematic [12] and three non-systematic [22,23,24]. In the systematic review, articles examining heat alerts in North America and Europe were analyzed. The focus of the non-systematic reviews was on heatwaves. One of the reviews discussed the subject globally [24], one reviewed literature on North America [22], and one focused on the U.S. [23]. As regards cold weather, there was one systematic review [25] with a global focus on economically developed countries. Additionally, one non-systematic review addressed both extreme heat and cold events, concentrating on Detroit in the U.S., and drew on literature globally for cities with similar climates [18].

#### 3.3.1. Heat Alerts, Heatwaves and Cold Weather

Reviews indicated that the way heat alerts are currently being communicated may not be reaching the most vulnerable populations, including homeless populations. Bassil and Cole, 2010 [12] found that there is generally an “uncertainty of whether public health messages actually reach the most vulnerable” (p. 997). Further, messages in the media tend to focus on specific sub-populations, such as those employed in outdoor occupations and children/pets left in cars. Those who do not belong to these sub-populations tend to believe that they do not belong to vulnerable groups and thus may be less likely to take the necessary precautions even when the heat alerts do reach them [12].

The systematic review on cold weather did not address homelessness directly. However, Tanner et al., 2013 [25] conclude that “low income, aspects of housing conditions and composite measures of fuel poverty were most consistently associated with mortality, morbidities or wider social outcomes in cold weather” (p. 1065). The housing conditions included such aspects as housing quality as well as poor thermal insulation and heating. Thus, the conclusions of this review may be relevant for homeless populations as well.

Gronlund et al., 2018 [18] note that there is “a higher risk of hypothermia, hyperthermia, and mortality among the homeless vs. non-homeless during periods of extreme heat or cold” (p. 56). Chronic diseases, the abuse of substances and alcohol, and mental health issues increase the vulnerability of homeless populations to heat and cold [18]. The risks associated with exposure to cold weather have not been discussed in non-systematic reviews. The increased vulnerability of homeless and low SES groups to extreme heat events has been linked to a higher prevalence of risk factors, such as physical and mental health conditions, among these populations [22,23]. The lack of access to resources necessary to address health issues further contributes to vulnerability [23]. Moreover, “up to 91% of homeless populations in the U.S. live in urban or suburban areas, where they are at increased risk from heat waves due to the heat island effect” [22] (p. 656). Shonkoff et al., 2011 [23] further indicate that there is less tree canopy coverage in neighborhoods with an increased proportion of residents living in poverty as well as with an increased proportion of racialized residents. Thus, the lack of green spaces exacerbates the effects of climate change-induced heatwaves. Although the available reviews focused on North America, it has been noted that vulnerable populations within low and middle-income countries may be at greater risk [24].

#### 3.3.2. Policy Recommendations and Further Research

In terms of policy recommendations, multiple approaches to addressing heatwaves were proposed, “ranging from modifying the built environment to improving housing and building standards” [24] (p. 145). However, no specific approaches to these issues were identified in the reviewed articles. The issue of delivering heat alert messages to vulnerable groups remains of primary importance. Ramin and Svoboda, 2009 [22] identify an adaptation strategy that is already in use in some cities: Heat-Health Warning Systems (HHWS), which are activated based on meteorological criteria. When the HHWS is initiated, “homeless shelters are requested to remain open, cooling stations are identified, transit tokens are distributed to homeless and outreach programs are activated” [22] (p. 660). Gronlund et al., 2018 [18] also suggest drawing on existing strategies, such as the guidance from the US Centers for Disease Control and Prevention for cooling centers and Toronto’s extreme heat and cold response plans. They further propose “increasing the number of cooling and warming centers with emergency power generators to provide short-term temperature relief but also shelter in other natural disasters or civil emergencies” [18] (p. 57). 

Recommendations for future research include exploring the strategies for delivering targeted messages to homeless populations prior to an extreme heat event [12,23]. Bassil and Cole, 2010 [12] suggest that examining “heat–health risk perception is one way to develop more targeted and effective communication strategies” (p. 999). Research should also investigate the effectiveness of the existing strategies [12]. Finally, Ramin and Svoboda, 2009 [22] indicate that there is a need for systematic data collection regarding the health of homeless populations (both in terms of pre-existing conditions and health outcomes) and extreme heat events. Concerning the cold weather, Tanner et al., 2013 [25] indicate that research is needed from multiple regions around the world. They also suggest that future reviews should focus on qualitative and intervention studies.

### 3.4. Health 

There was no systematic review that focused specifically on the relationship between climate change and the health of homeless and vulnerably-housed groups. Jessel et al., 2019 [19], however, included a discussion of health in their review of energy insecurity. This review focused on the U.S. and the Global North. By contrast, six non-systematic reviews addressed the influence of climate change on the health of homeless and low-income populations. Four of the reviews discussed the subject in global terms [14,20,21,24], one focused on North America [22], and one concentrated on the U.S. [23].

#### 3.4.1. Projected Health Outcomes

In their discussion of energy insecurity, Jessel et al., 2019 [19] indicate that the negative health effects due to energy poverty are intensified by the changing climate. They further note that “stress from having to make trade-offs between basic needs for food, water, housing, and energy profoundly affects adult and child mental health” [19] (p. 10). Racial minorities and low-income families tend to be more exposed to environmental hazards, which leads to increased health risks. Further, a lack of access to air conditioning may exacerbate health problems during extreme heat events, as a fear of violence has been found to be a reason for not travelling to cooling centers and keeping the windows closed.

The non-systematic literature reviews primarily focused on projected health outcomes. Specifically, climate change is expected to adversely affect the health of homeless populations by increasing vulnerability to infectious and vector-borne diseases [14,20,21,22,24]. For example, Ramin and Svoboda, 2009 [22], in their discussion of the West Nile Virus, indicate that “an earlier onset of spring resulting from climate change is projected to prolong the amplification cycle of the virus resulting in an increased incidence of human infection” (p. 657), especially affecting homeless individuals as they tend to sleep outdoors. Homeless populations and low SES groups are also disproportionately affected by pre-existing conditions, further increasing their vulnerability [22,23,24]. In particular, the negative health effects of respiratory and cardiovascular conditions will be compounded by exposure to the climate change-induced increases in outdoor air pollution [22]. It is also anticipated that climate change will influence mental health adversely, possibly due to “personal experience of loss or injury from an extreme event, exposure to media coverage of these events, and stress and anxiety over potential future impacts” [21] (p. 546). Some research suggests that there may be positive health effects as well, mainly due to warmer winters, but such health benefits “will be minor and will be outweighed by the adverse health impacts of climate change” [22] (p. 658). Other health risks may include violence, particularly between the migrant and the host groups [14].

#### 3.4.2. Policy Recommendations and Further Research

Most recommendations provided in the reviews propose addressing issues related to health by providing adequate housing and implementing structural changes. These recommendations are discussed in the corresponding sections. However, Kjellstrom and Mercado, 2008 [20] found that health can be effectively used as a rallying point in introducing public policy: “health can unite individuals, communities, institutions, leaders, donors, and politicians, even in complex and hostile contexts where structural determinants of health are deep and divisive” (p. 566). The authors do not provide further details on implementing this approach. Reviews commented on the need for research focusing on the relationship between climate change and the health of low-income and homeless populations [22,24], with particular attention to structural challenges and racial inequalities [19,23]. Further, it was suggested that research should explore the individual and structural responses that enable resilience [21,24]. Finally, Jessel et al., 2019 [19] suggest “studying the energy–health–justice nexus through the lens of acute and chronic energy insecurity” (p.14), as energy insecurity increases vulnerability to climate change, which can have detrimental effects on the health of a population.

### 3.5. Structural Factors

Structural factors as they relate to homelessness and climate change were discussed in five reviews: one systematic [26] and four non-systematic [14,15,20,22]. The systematic review addressed green infrastructure in urban planning and drew on global research with a focus on the applicability of green infrastructure in sub-Saharan Africa. The non-systematic reviews focused mostly on recommendations for structural changes. Three of the reviews discussed this in global terms [14,15,20], while one focused on North America [22].

#### 3.5.1. Green Infrastructure

Titz and Chiotha, 2019 [26] discuss green infrastructure in urban environments as a response to the impacts of climate change, which disproportionately influence marginalized communities. Although green infrastructure can have “beneficial bio-physical as well as social effects” [26] (p. 12), the authors express concern over its uneven distribution and potential to increase social inequality. In particular, Titz and Chiotha, 2019 [26] indicate that the disadvantaged groups have less access to new green spaces, and bringing green infrastructure into poorer neighborhoods can even lead to “paradoxical effects, as rising housing costs and property values may result in ecogentrification” [26] (p. 13). Given the substantial benefits that can be derived from green infrastructure, Titz and Chiotha, 2019 [26] highlight the importance of socially inclusive planning.

The discussion of structural factors as they relate to climate change and homelessness is limited in the non-systematic literature reviews. Costello et al., 2009 [14], for instance, indicate that the local government should take responsibility for providing protective infrastructure for vulnerable groups, both urban and rural. The authors further note the importance of recognizing women as actors of change, especially in developing countries, where women tend to be one of the groups most vulnerable to climate change [14]. Many of the non-systematic reviews provide recommendations and suggestions for further research to improve the response to climate change.

#### 3.5.2. Policy Recommendations and Further Research

Many of the reviews recommend focusing on land-use and urban planning. Based on their review, Titz and Chiotha, 2019 [26] conclude that in city planning, more focus should be placed on green infrastructure and particularly on urban agriculture. Costello et al., 2009 [14] highlight the importance of technological adaptations and of providing low-cost alternatives to ensure accessibility. Specifically, in relation to extreme events, they suggest focusing on “improvements in regional and local climate modelling, development of effective early warning systems, and application of the geographic information system to improve vulnerability assessment, hazard and risk zonation, and land-use planning” [14] (p. 1718). The authors also indicate the need for policies aimed at empowering women [14]. Further, Ramin and Svoboda, 2009 [22] suggest concentrating on policies that reduce air pollutants, including “alternative energy policies, improved transport systems, improved urban planning, and carbon markets” (p. 660).

Kjellstrom and Mercado, 2008 [20] indicated that high-income countries should provide funding to low-income countries to help them to address the structural challenges that they face. However, the authors caution that increased funding “would have little impact without a very considerable strengthening of local governance capacity for good health care, emergency services and environmental health, and a greater willingness by local government to work with low-income groups” [20] (p. 566). They do not provide specific suggestions on how to address the identified issues. Finally, Dodman, 2009 [15] addressed the issue of population density and concluded that de-densification might cause an increase in greenhouse gas emissions. Instead, it is suggested in this review that one approach that has been successfully implemented is to “make adequate and appropriately located land available to low-income urban groups” [15] (p. 72).

Titz and Chiotha, 2019 [26] suggest that, in relation to green infrastructure, further research is needed that focuses on rights-based approaches to address the inequalities arising from urban space planning. They also note that most research on green infrastructure has been conducted in European cities and indicate a need to examine “to what extent these concepts can be simply transferred to the heterogeneous urban development processes in sub-Saharan Africa” [26] (p. 18).

### 3.6. Natural Disasters

Natural disasters were discussed in three non-systematic reviews [16,17,22]. No systematic reviews on the subject were identified. Fothergill and Peek, 2004 [16] focus on the relationship between poverty and natural disasters but do not refer to climate change. The other two reviews [17,22] look specifically into homelessness and climate change. The reviews concentrate on North America [22] and the U.S. [16,17].

#### 3.6.1. Exposure

Natural disasters increase the vulnerability of homeless populations as they tend to be more exposed to the risks [22] and as the demands for shelters may exceed their capacity [16,17]. Ramin and Svoboda, 2009 [22] found that disaster planning frequently overlooks urban homeless populations: “no adaptation measures specific to the homeless are reported to be in place for… floods and storms” (p. 660). Further, as a result of natural disasters, there tends to be an increase in homelessness due to displacement [16] and due to natural disasters impeding the efforts of those recovering from homelessness [17]. Fothergill and Peek, 2004 [16] also note that people with a low SES are “more likely to be financially devastated by the disaster and subsequent relocation” (p. 96). Those who are more affluent, such as middle and higher SES groups, tend to have more financial resources as well as social support networks to help them to adjust following a natural disaster. Further, homeless and low-income women experience greater difficulties with relocating to shelters following disasters and have more limited opportunities to recover their material losses [16].

#### 3.6.2. Recommendations and Further Research

Gibson, 2019 [17] recommends establishing relationships with homeless population members to help locate and engage homeless individuals when natural disasters occur. Further, individuals experiencing homelessness should be provided information regarding shelters, including their location and transportation options, in advance [17]. Shelters also should be accessible, and emergency transportation, such as shuttles, should be available [17]. Gibson, 2019 [17] further indicates that organizations working with homeless people “may need to educate and advocate on behalf of this population to other community entities to help ensure they are not denied access to available resources” (p. 5). Homeless populations also need to be considered in disaster planning [17]. The availability of affordable and safe housing is also emphasized [16,17]. The suggested strategies are discussed in the corresponding section.

The reviews comment on the need for in-depth, comparative studies looking into vulnerability issues, including diversity within homeless populations and problems that emerge during evacuations (such as potential increase in domestic violence), and into the effects of different natural disasters [16] as well as the integration of practices based on the available evidence [17]. Fothergill and Peek, 2004 [16] suggest that those working in communities should be consulted in the process of research development. Research should also focus on “how low-income communities perceive risk, prepare for disasters, and respond to warning communications” [16] (p. 105). Gibson, 2019 [17] suggests that clear and consistent definitions of the terms “sheltering” and “housing” in literature focused on disasters should be developed.

### 3.7. Housing

The availability and quality of housing remain the most prominent concerns in supporting homeless and marginally-housed populations. These issues are addressed in four reviews: one systematic [13] and three non-systematic [15,16,17]. The systematic review looks at Latin American, African, and Asian cities. One non-systematic review discusses the issue in global terms [15], while the other two are focused on the U.S. [16,17]. The reviews offer limited information on findings and further research suggestions in this area. Therefore, the current section focuses on provided recommendations.

Regarding quality, some reviews emphasize the need for upgrading urban slums, with more successful strategies being based on state–community interaction [13,15]. Corburn and Sverdlik, 2017 [13] recommend relying on existing practices, such as Health Impact Assessment and Health in All Policies frameworks, as they “may help further elucidate how slum upgrading projects can be designed and evaluated for their potential health impacts” (p. 7). Dodman, 2009 [15] suggests that new low-cost housing should be designed with more attention paid to the existing strategies employed by the urban poor to adapt their dwellings to the climatic demands of their regions. Further, regarding rental housing, Fothergill and Peek, 2004 [16] suggest implementing city-based rent control policies as well as mandating or providing subsidies to landlords to improve low-cost rental housing quality. Finally, Gibson, 2019 [17] recommends finding “strategies that better utilize existing government subsidies” (p. 4) to increase the availability of low-cost housing.

## 4. Discussion

### 4.1. Key Findings

The results of this systematic review of reviews highlight the paucity of research syntheses addressing the effects of climate change and weather extremes on homeless and vulnerably housed populations. This is a major global challenge and, while issues such as climate-driven migration are receiving substantial attention, the implications for housing loss and instability have not been substantially addressed. Literature syntheses, specifically in this area, will be essential for developing coherent, transferable, and relevant response strategies. This need for review-level information is particularly pertinent for problems with diverse and complex issues and contexts [27], as is the case with the climate–homelessness nexus.

Despite the limited number of reviews identified in the present review, the intent to identify systems considerations and recommendations was addressed. These reviews suggest multiple interconnected pathways in which climate change and weather extremes influence homelessness. First, there is an increased risk of homelessness for precariously-housed populations due to energy insecurity [19] and due to climate change-induced natural events, which may destroy or worsen housing conditions [23]. Unsurprisingly, there is evidence that homeless and vulnerably-housed populations are disproportionately exposed to climatic events, including extreme heat, cold, and natural disasters. During extreme heat events, public health messages may not have the capacity to reach vulnerable populations [12]. Further, homeless populations tend to be excluded in disaster planning, exacerbating their exposure [17]. Lower SES groups also tend to lack the financial and social resources needed to recover from natural disasters, especially those leading to relocation [16]. There are two additional risk domains in this area that might be less obvious but that emerged in the literature: the influence of weather extremes and climate change on the incidences of infectious and vector-borne diseases amongst homeless populations, who are both more exposed to infection and more vulnerable due to pre-existing conditions [22]; and mental health issues, which are also expected to worsen, mainly due to the stress associated with extreme climatic events [21]. The depth to which these themes were addressed varied, with some areas (e.g., air pollution and the interacting risks of exposure to cold weather conditions) clearly needing more attention.

### 4.2. Human Rights-Based Approach (HRBA)

Given the wide range of interventions found in this review of reviews (see Table 2), there is a need for a framework to better understand how to organize policy, planning, and practice responses. A key re-emerging theme in the recommendations provided in the reviews is the need to address systemic inequalities faced by homeless and marginally housed populations, especially in the context of climate change. The HRBA could be useful as an integrative framework to help organize housing-related responses to climate change. Within the HRBA framework, extreme poverty is considered to be a violation of human rights. Consequently, those that are able to provide support are not seen as giving charity, but as having an “obligation to respect, protect and fulfill rights” [28] (Chapter 2) and are thus defined as duty bearers, whereas the beneficiaries are rights holders. The HRBA framework is particularly useful in the design and implementation of projects tailored to the demands of a given context. For example, Hearne and Kenna, 2014 [29] applied the HRBA to the issue of housing deprivation in an Irish urban housing estate. Teaching tenants and community workers about human rights enabled the community to identify key areas in which their human rights were violated. Next, specific indicators were developed to help assess the situation and to monitor progress. After gathering evidence, public human rights hearings were organized. Finally, duty bearers were engaged in the process formally, and a monitoring process was established.

The HRBA framework could be applied across a range of contexts where climate change is impacting or will impact homeless and marginally housed populations. The recommendations by Kjellstrom and Mercado, 2008 [20], for instance, can be adapted as follows: high-income countries (duty bearers) should provide funding to low-income countries (rights holders) to ensure the availability of necessary resources. Low-income countries, in turn, should act as duty bearers towards their citizens (rights holders), ensuring the appropriate use of funds.

## 5. Limitations

This review of reviews presents a synthesis of the current knowledge on the influence of climate change on homeless and vulnerably-housed populations—a prominent topic that needs to be addressed urgently. While a review-level analysis provided greater systems level guidance and an increased breadth of the topic, several limitations are noteworthy. First, a limited number of reviews was identified, only some of which were systematic, potentially signaling a need to consider other key terms related to housing challenges. As a result, some of the reviews included did not focus exclusively on climate change and homeless and marginally-housed populations. Second, due to the scarcity of data available on the subject, some reviews (especially those concerning housing and structural conditions) provided recommendations that did not rely on data analyses. Some of the recommendations also lack clear suggestions for implementation (e.g., ensuring intended use of funds transferred to low-income countries). Third, there is a lack of research available on low-income countries, where the problems associated with climate change and housing are most prominent. Fourth, limited attention is given to the exacerbated disadvantages experienced by the most vulnerable groups: indigenous peoples, racialized minorities, women, and migrants. This gap in research mirrors the shortcomings at the original research level identified in the scoping review [7]. Therefore, more studies are needed to confirm the results presented here and provide a more nuanced understanding of the issues.

## 6. Conclusions

Climate change and associated weather extremes present significant and immediate risks for populations lacking shelter. Moreover, in this review of reviews, we found that climate change-related events seem to contribute to the prevalence of homelessness through migration, poverty, and other intersecting stressors. These problems are likely to become worse as the climate change emergency worsens, but there is a lack of reliable data syntheses of these risks and how they are unfolding, which hampers prevention and crisis response planning, policy development, and risk modeling. Thus, there is a need for the further identification and integration of research related to homelessness and housing precarity. However, the reviewed literature suggests that efforts to intervene should concentrate on systemic responses to inadequate housing and lack of shelter. Prevention-oriented work, in most contexts, will need to occur alongside crisis response activities, given the large and growing number of individuals displaced by weather extremes and exposed to the elements while experiencing homelessness and compromised health. Inclusion and equity in crisis response will require advocacy and education in most contexts where homeless individuals are not considered in disaster planning and other risk mitigation efforts (e.g., green urban infrastructure). Planning and implementation should involve close collaboration with direct service providers and individuals with lived experience to develop effective ways of engaging these marginalized populations and implementing strategies that are relevant to a given context. While activities targeting homeless and at-risk populations will be essential to reducing mortality and morbidity associated with climate change in this population, it will be crucial to integrate this work into broader risk mitigation and response efforts. Considering homelessness as a somehow separate entity from overall policy and planning considerations will likely lead to ongoing stigmatization, inefficiencies, and reduced effectiveness.

## Figures and Tables

**Figure 1 ijerph-18-05812-f001:**
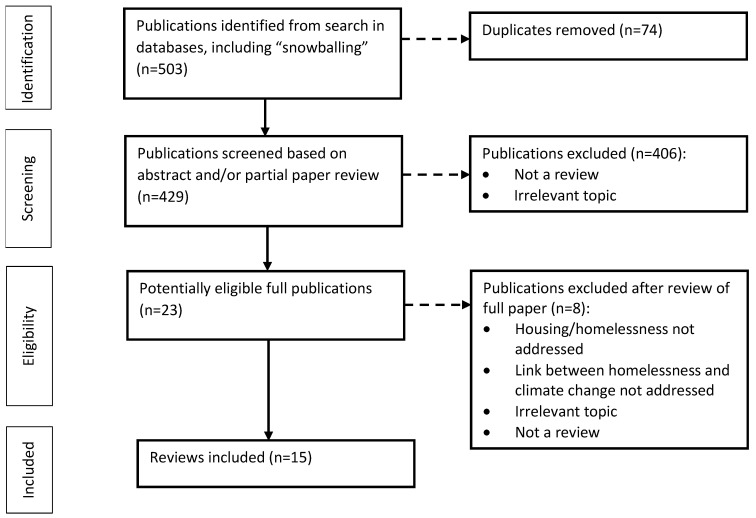
Process of data evaluation and study selection.

**Table 1 ijerph-18-05812-t001:** Included reviews: characteristics and identified themes.

Authors	Year	Focus Areas	Type of Review	Resources Included	Themes	Relevant Results and Recommendations
Bassil & Cole [12]	2010	U.K., Canada, U.S., France, Czech Republic, Portugal	Systematic: structured review	14 records: peer-reviewed articles and grey literature	Heat and cold	Unclear if heat alerts reach the vulnerable populations
Coburn & Sverdlik [13]	2017	Latin American, African and Asian cities	Systematic: structured review	19 projects; 39 records: peer-reviewed articles and grey literature	Housing	Slum upgrading initiatives more successful when based on state-community interaction
Costello et al. [14]	2009	Global	Non-systematic	N/A	Health	Lack of resources (especially housing and water) increases vulnerability to infectious and vector-borne diseases
					Structural factors	Housing issues of those living in poverty intensify due to climate change; they should be addressed by the local government
Dodman [15]	2009	Global	Non-systematic: analytical review	N/A	Structural factors	Greenhouse gas emissions may increase as a result of de-densification policies
					Housing	State-community interaction important for successful slum upgrading; local dwellers’ knowledge should be used in housing development
Fothergill & Peek [16]	2004	U.S.	Non-systematic: literature review	N/A	Natural disasters	Shelter capacity can be exceeded; displacement increases homelessness; low SES more effected
					Housing	Low-income housing quality and affordability should be addressed by local government
Gibson [17]	2019	U.S.	Non-systematic: literature review	N/A	Natural disasters	Resources of organizations providing support strained; hamper the efforts of those recovering from homelessness
					Housing	Increase availability of affordable housing by implementing more cost-effective strategies
Gronlund et al. [18]	2018	U.S.	Non-systematic: narrative review	N/A	Heat and cold	Health and lives of homeless populations at increased risk during extreme weather events
Jessel et al. [19]	2019	U.S. and Global North	Systematic: literature review	162 records: books, peer-reviewed articles, reports	Risk factors	Energy insecurity is exacerbated by climate change; contributes to increase in homelessness due to evictions
					Health	Energy insecurity has negative direct and indirect impacts on health; adverse effect on mental health
Kjellstrom & Mercado [20]	2008	Global	Non-systematic	N/A	Risk factors	It is projected that increasing numbers of people will become environmental refugees and end up in slums due to climatic events
					Health	Negative health effects of climate change are projected to result from climatic events and increase in diseases
					Structural factors	Addressing structural changes requires funding from high-income to low-income countries
Leichenko & Silva [21]	2014	Global	Non-systematic: literature review	N/A	Risk factors	Migration due to climatic events from rural areas to urban slums; the poorest might not have resources to migrate
					Health	Expected increase in diseases effecting the poor; mental health is undermined by loss, relocation, stress, and anxiety
Ramin & Svoboda [22]	2009	North America	Non-systematic: literature review	N/A	Heat and cold	Increased vulnerability to heat due to preexisting health conditions; more exposed to heat island effect in urban and suburban areas
					Health	Increase in infectious and vector-borne diseases, exacerbated by preexisting health conditions; minor positive impacts of climate change
					Structural factors	To mitigate health effects on homeless populations, more policies aimed at reducing air pollution should be implemented
					Natural disasters	Increased vulnerability of the homeless populations; lack of inclusivity in disaster planning
Shonkoff et al. [23]	2011	U.S. (California)	Non-systematic: literature review	N/A	Risk factors	Vulnerably housed lack resources to prepare, respond and recover from climatic events
					Heat and cold	Low SES groups effected most adversely by extreme heat events; exacerbated by lack of access to resources
					Health	Preexisting health conditions increase vulnerability
Sverdlik [24]	2011	Global	Non-systematic: literature review	N/A	Heat and cold	Vulnerable populations in low- and middle-income countries at increased risk from heatwaves
					Health	Anticipated increase in diseases, exacerbated by preexisting health conditions
Tanner et al. [25]	2013	Global (economically developed countries)	Systematic: narrative approach	33 records: 1 systematic review; 19 individual-level studies; 13 studies using ecological data	Heat and cold	Precariously housed low SES groups are more exposed to health risks associated with cold weather and also experience negative social outcomes
Titz and Chiotha [26]	2019	Global (focus on Africa)	Systematic: literature review	Total records unspecified: peer-reviewed articles and book chapters	Structural factors	Green infrastructure as a response to climate change; issues of ecogentrification should be addressed by inclusivity in urban planning

**Table 2 ijerph-18-05812-t002:** Interventions.

**Homeless and vulnerably-housed populations and weather extremes:** Offer outreach programs to establish relationships with homeless populations Provide information on shelters to homeless populations in advance Ensure shelter accessibility (e.g., shuttle buses in emergency situations) Examine heat-health risk perceptions Educate community entities on local homeless populations to ensure homeless people have access to the available resources Draw on existing practices in relation to weather extremes (e.g., the guidance from the US Centers for Disease Control and Prevention for cooling centers; Toronto’s extreme heat and cold response plans; Heat–Health Warning Systems) Increase number of cooling and warming centers
**Housing and Urban planning:** Rely on existing practices to upgrade urban slums (e.g., Health Impact Assessment and Health in All Policies frameworks) Design low-cost housing by relying on knowledge of local dwellers Establish clear and consistent definitions of the terms “sheltering” and “housing” in literature focused on disasters Implement city-based rent control policies as well as mandate or provide subsidies to landlords to improve low-cost rental housing quality Avoid de-densification, land to low-income urban groups instead Focus on green infrastructure in urban planning Pursue research focused on rights-based approaches and in varying contexts Include homeless populations in disaster planning
**Research:** Examine the relationship between climate change, energy insecurity, and health from the perspective of the energy–health–justice nexus Identify racial and gender-based injustices Consult those working in communities during research development Cover a wider range of countries Focus on systematic data collection and on intervention, with qualitative and in-depth comparative studies Evaluate the effectiveness of existing strategies Integrate practices based on the available evidence Examine individual and structural responses that enable resilience
**Other:** Commit funds from high to low-income countries Ensure local governments’ willingness to work with low-income groups Use technology for climate modeling, the development of warning systems, and vulnerability zonation Use “health” as a rallying point

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
