# Peer review of "Climate Change, Weather, Housing Precarity, and Homelessness: A Systematic Review of Reviews"

_ijerph, 2021, doi:10.3390/ijerph18115812_

Round 1
Reviewer 1 Report
This paper employs thematic analysis to examine housing precarity and homelessness in relation to climate change and weather extremes. Overall, this paper is easy to follow; however, I have concerns on the research design of this paper. Firstly, it is not normal to review the existing reviews. Especially, there are only 15 reviews can be investigated and the reviews are from 2004 to 2019, I think it is not scientific to obtain sufficient reliable findings from only reviewing 15 reviews. Secondly, the keywords used for searching existing papers are controversial. There are many other keywords which can represent similar literal meanings. Thus, there is a great possibility that this study does not include all of the relevant existing reviews.
Reviewer 2 Report
This is a very good review on the topic of the impact of climate change on housing precarity and homelessness. I was just surprised that the topic air pollution was excluded in the review. Otherwise, the paper is publishable in its present form.
Reviewer 3 Report
A paper that needs just a few suggested dentitions to be clarified for the potential readers. These would include the description of “ snowballing” as used here( line 113), and define systemic and non- systemic reviews as used here( line 127). Two other thoughts not especially needed but nice. Mention tree canopy in cities as mitigating heat islands which are generally more common in poorer areas, and some discussion of the feasibility of some suggestions made, such as fund transfers to poorer countries and the likelihood that such funds would actually be used as intended. On lines 200 and 204 this reviewer has some concerns that the concept of increased morbidity from cold is somewhat speculative and not supported in the same way as exists for heat increases. If there is excess mortality does it come strictly from the cold or associated other factors such as alcohol abuse.
